# Developing adapted wheat lines with broad-spectrum resistance to stem rust: Introgression of *Sr59* through backcrossing and selections based on genotyping-by-sequencing data

Mahboobeh Yazdani[1], Matthew N. Rouse[2,3], Brian J. Steffenson[3], Prabin Bajgain[4], Mehran Patpour[5], Eva Johansson[1], Mahbubjon Rahmatov[1] *

1 Department of Plant Breeding, Swedish University of Agricultural Sciences, Alnarp, Sweden, 2 United States Department of Agriculture, Agricultural Research Service, Cereal Disease Laboratory, St. Paul, MN, United States of America, 3 Department of Plant Pathology, University of Minnesota, St. Paul, MN, United States of America, 4 Department of Agronomy and Plant Genetics, University of Minnesota, St. Paul, MN, United States of America, 5 Department of Agroecology, Aarhus University, Slagelse, Denmark

* Mahbubjon.Rahmatov@slu.se

**Data Availability Statement:** All relevant data are within the paper.

## Abstract

Control of stem rust, caused by *Puccinia graminis* f.sp. *tritici*, a highly destructive fungal disease of wheat, faces continuous challenges from emergence of new virulent races across wheat-growing continents. Using combinations of broad-spectrum resistance genes could impart durable stem rust resistance. This study attempted transfer of *Sr59* resistance gene from line TA5094 (developed through CSph1bM-induced T2DS·2RL Robertsonian translocation conferring broad-spectrum resistance). Poor agronomic performance of line TA5094 necessitates *Sr59* transfer to adapted genetic backgrounds and utility evaluations for wheat improvement. Based on combined stem rust seedling and molecular analyses, 2070 $BC_1F_1$ and 1230 $BC_2F_1$ plants were derived from backcrossing BAJ#1, KACHU#1, and REEDLING#1 with TA5094. Genotyping-by-sequencing (GBS) results revealed the physical positions of 15,116 SNPs on chromosome 2R. The adapted genotypes used for backcrossing were found not to possess broad-spectrum resistance to selected stem rust races, whereas *Sr59*-containing line TA5094 showed resistance to all races tested. Stem rust seedling assays combined with kompetitive allele-specific PCR (KASP) marker analysis successfully selected and generated the $BC_2F_2$ population, which contained the *Sr59* gene, as confirmed by GBS. Early-generation data from backcrossing suggested deviations from the 3:1 segregation, suggesting that multiple genes may contribute to *Sr59* resistance reactions. Using GBS marker data (40,584 SNPs in wheat chromosomes) to transfer the recurrent parent background to later-generation populations resulted in average genome recovery of 71.2% in BAJ#1*2/TA5094, 69.8% in KACHU#1*2/TA5094, and 70.5% in REEDLING#1*2/ TA5094 populations. GBS data verified stable *Sr59* introgression in $BC_2F_2$ populations, as evidenced by presence of the *Ph1* locus and absence of the 50,936,209 bp deletion in CSph1bM. Combining phenotypic selections, stem rust seedling assays, KASP markers,

**Funding:** For this research financial support obtained from the Swedish Research Council Vetenskapsrådet and FORMAS. There was no involvement of the funding agencies in the study's design, the collection and analysis of data, the decision to publish, or the preparation of the manuscript.

**Competing interests:** There are no competing interests among the author.

and GBS data substantially accelerated transfer of broad-spectrum resistance into adapted genotypes. Thus, this study demonstrated that the *Sr59* resistance gene can be introduced into elite genetic backgrounds to mitigate stem rust-related yield losses.

## Introduction

Wheat (*Triticum aestivum* L.) is an important source of calories and protein in the daily human diet world-wide [1]. Due to the current rapid growth in the global population, a 60% increase in wheat production will be necessary in order to maintain its current share of the human diet by 2050 [2]. Wheat yield will need to be increased by at least 2% each year to meet this demand, a target that is currently not being attained [3]. The major constraints to achieving the necessary yield increase are biotic and abiotic stresses that impair crop performance, with rust diseases in particular having the potential to cause yield losses in severe outbreaks. Among these diseases, stem rust (caused by the fungus *Puccinia graminis* f. sp. *tritici* (*Pgt*)) is a major threat to wheat production across many regions of the world, because it is capable of causing severe yield loss [4]. Although fungicide application can effectively manage stem rust, it is associated with drawbacks such as high costs, significant environmental impact, and negative effects on human health [5]. Hence, genetic resistance is the most economical and environmentally sustainable control measure to protect wheat yields from the threat of stem rust. The frequent emergence of new *Pgt* races is a major challenge to success in breeding resistance to this pathogen in wheat. An example of this is emergence of the Ug99 race group, which is capable of overcoming all known and extensively deployed stem rust (*Sr*) resistance genes, including *Sr24*, *Sr31*, *Sr36*, and *SrTmp*. This constant adaptation of the pathogen has increased concerns about global epidemics [6, 7].

Other widely virulent *Pgt* races, such as TRTTF, TKTTF, TTRTF, TTKST, PRCTM, and TTKTT, have been found to possess additional virulence combinations, including virulence to *Sr22+Sr24*, *Sr24+Sr31*, *Sr13b+Sr35+Sr37*, and *Sr24+Sr31+SrTmp* genes [8–10]. The emergence of these novel races and their spread into Europe is alarming, since stem rust disease has largely been absent for nearly 60 years [11–13]. Moreover, a high proportion of cultivars grown in Europe are susceptible to these emerging races, e.g., ~80% of wheat cultivars currently grown in the United Kingdom are susceptible to race TKTTF [11], while resistance genes such as *Sr24*, *Sr31*, and *Sr38*, present in German wheat cultivars are limited in their effectiveness against these novel races of *Pgt* [14]. Of the currently known and described wheat *Sr* genes, 35 out of 73 derive from the primary gene pool of wheat and the majority of these do not confer broad-spectrum resistance [4]. Until recently, the *Sr31* resistance gene was considered highly effective in conferring broad-spectrum resistance against all *Pgt* races, and was introduced into many wheat cultivars over the past 30 years [7].

Rye (*Secale cereale* L., 2n = 14), belonging to the tertiary gene pool of wheat, is an important source of genes that can be used for increasing bread wheat resistance to both abiotic and biotic stresses [15]. For instance, *Sr27*, *Sr31*, *Sr1RS*[Amigo], *SrSatu*, *Sr50*, and *Sr59* are important stem rust resistance genes that have been introduced into wheat from rye, and several of these genes have been proven to confer broad-spectrum resistance [16–19]. However, introgression of genes from wild relatives into wheat relies on meiotic recombination, which is complicated between rye and bread wheat. Hexaploid bread wheat (*Triticum aestivum* L., 2n = 6x = 42, AABBDD) was derived from *T. urartu* (2n = 2x = 14, AA), *Aegilops sp.* (2n = 2x = 14, BB), and *Ae. tauschii* (2n = 2x = 14, DD) through spontaneous interspecific crosses. Allohexaploid wheat behaves as a diploid during meiosis [20], due to the presence of *pairing homoeologous*

(*Ph*) loci that strictly control pairing homology during meiosis. Two major *Ph* loci (*Ph1* and *Ph2*, residing on chromosome 5BL and 3DS, respectively) control homoeologous recombination in wheat [20, 21]. Deletion of the *Ph1* locus (*ph1b* in hexaploid and *ph1c* in tetraploid wheat) results in homoeologous recombination [22, 23]. As a result, a Chinese Spring line mutated at the *ph1b* locus (CSph1bM) has been used effectively to induce recombination between wheat and alien chromosomes [24]. Many broad-spectrum resistance genes have been transferred using CSph1bM, including *Sr32* from *Ae. speltoides* [24], *Sr39* from *Ae. speltoides* [25], *Sr47* from *Ae. speltoides* [26], *Sr53* from *Ae. geniculata* [27], *Sr43* from *Thinopyrum ponticum* [28], *Sr47* from *Ae. speltoides* [26], *Sr53* from *Ae. geniculata* [27], *Sr59* from *S. cereale* [16], and *Yr83* from *S. cereale* [29]. Line CSph1bM has also been used to transfer genes other than those for wheat resistance, e.g., end-use quality has been improved by recombining the *Sec-1* (secalin) allele on the 1RS chromosome arm in wheat lines [30].

A large number of wheat-rye introgression lines were developed in the 1980s-2000s by the late Professor Arnulf Merker at the Swedish University of Agricultural Sciences [31, 32]. Some of these lines were used in field and greenhouse screenings to identify the line 'SLU238' [2R (2D) wheat-rye disomic substitution], which was found to confer broad-spectrum resistance to all *Pgt* races tested [33]. TA5094, a line derived from 'SLU238', has since been shown to possess a T2DS·2RL Robertsonian translocation with a stem rust resistance gene designated *Sr59* [16]. Due to the lack of acceptable agronomic performance in TA5094 based on the CSph1bM background, there is an urgent need to transfer this gene to a more suitable genetic background and evaluate its potential use in wheat resistance breeding. This paper describes transfer and subsequent evaluation of *Sr59* to agronomically suitable genetic background derived lines through: 1) marker-assisted backcross breeding; 2) stem rust seedling assessment; 3) background selection; and 4) physical mapping of the *Sr59* resistance gene on chromosome 2RL.

## Materials and method

### Plant materials and stem rust seedling evaluations in parental lines

TA5094 was derived from a cross between CSph1bM and line SLU238 [a 2R (2D) wheat-rye disomic substitution], and has been defined as a T2DS·2RL translocation containing the *Sr59* resistance gene [16]. In the present study, three spring bread wheat cultivars (BAJ#1, KACHU#1, and REEDLING#1), kindly provided by Dr. Ravi Singh (International Maize and Wheat Improvement Center (CIMMYT), El Batan, Mexico), were used as recurrent parents and crossed with TA5094. The resulting progeny were evaluated by seedling tests for stem rust reaction, molecular marker analysis, kompetitive allele specific PCR [KASP] markers, and genotyping by sequencing [GBS], with the four lines TA5094, CSph1bM, SLU238, and Chinese Spring (CSA) used as controls. The selected parental lines and controls were initially tested with the *Pgt* races TTTTF (isolate 01MN84A-1-2), TTTTF (isolate RU118b/16), QTHJC (C25; isolate 1541), TPMKC (C53; isolate 1373), RKQQC (C35; isolate 1312), RCRSC (isolate 77ND82A), TTRTF (isolate IT14a/16), TKTTF (isolate 13ETH60), TKTTF (isolate IQ115a/14 and isolate SE27121), TTKTT (isolate 14KEN58-1), TTKSK (isolate 04KEN156/04), TTKST (isolate 06KEN19v3), TTTSK (isolate 07KEN24-4), JRCQC (08ETH03-1), TRTTF (isolate 06YEM34-1), and LTBDC (Australian *Pgt* race 98–1,2,3,5,6).

### Population development, stem rust seedling evaluations, and molecular marker analysis

The $F_1$ plants obtained from crosses between line TA5094 and the recurrent parents (BAJ#1, KACHU#1, and REEDLING#1) were backcrossed to each of the corresponding recurrent

parents, generating $BC_1F_1$ seeds. A total of 2,070 $BC_1F_1$ plants were assessed for their seedling responses to *Pgt* race TTTTF (isolate 01MN84A-1-2), in trials at the USDA-ARS Cereal Disease Laboratory and University of Minnesota using a previously described stem rust seedling assay [34, 35]. For each recurrent parent, 94 resistant $BC_1F_1$ plants were selected (i.e., in total 282 plants) and analyzed for the presence of *Sr59* by use of three KASP markers: KASP_2RL_c25837C1, KASP_2RL_c21825C1, and KASP_2RL_c20194C2 [16]. Based on the results of KASP marker analysis, $BC_1F_1$ plants with *Sr59* were selected and used for backcrossing to produce $BC_2F_1$ plants. The backcross generated 1,230 $BC_2F_1$ plants, which were assessed against *Pgt* race TTTTF. Resistant plants were selected and tested for the presence of *Sr59* with the three KASP markers. Plants carrying *Sr59* were selfed to produce the $BC_2F_2$ generation, which resulted in a total of 846 families (from all recurrent parents). These families were again evaluated against *Pgt* race TTTTF and resistant plants were selected and checked with the three KASP markers. The $BC_2F_2$ generation was also evaluated using race TTKSK, and the pattern of segregation was analyzed. From the $BC_2F_2$, additional generations ($BC_2F_3$, $BC_2F_4$, and $BC_2F_5$) were created through selfing, 20 plants from each generation were selected based on the seedling response to race TTTTF, and presence of *Sr59* was validated by KASP markers. In addition, 10–15 $BC_2F_4$ and $BC_2F_5$ plants from each family were tested for their seedling response to races TTKSK, TTTSK, and TRTTF. The $BC_2F_5$ families were also assessed against races TPMKC, QTHJC, RKQQC, and RCRSC. The segregation pattern data were assessed using chi-square ($\chi^2$) analysis.

## Genotyping and data analysis

The population was genotyped using GBS as described previously [36]. Tissue sampling and DNA extraction were carried as described previously [37]. Approximately 10 cm of young leaf tissue from each of the donor parents (TA5094 and CSph1bM), recurrent parents, and a total of 128 selected $BC_2F_2$ plants (from all three recurrent parents) were collected in a 96-well tissue collection plate. Genomic DNA was isolated using the Qiagen BioSprint 96 instrument and the associated Qiagen BioSprint DNA Plant kit (https://www.qiagen.com/us/products/discovery-and-translational-research/dna-rna-purification/dna-purification/genomic-dna/biosprint-96-dna-plant-kit/#orderinginformation). DNA sequencing libraries were prepared and sequenced at the University of Minnesota Genomics Center. In brief, the isolated DNA was quantified with PicoGreen for GBS genotyping and normalized to 20 ng/μL. The GBS libraries were prepared in 96-plex using two restriction enzymes: a rare cutter *Pst*I (5′– CTGCAG–3′) and a frequent cutter *Msp*I (5′–CCGG–3′) with a common reverse adapter ligated [36, 38]. Libraries were sequenced on Illumina HiSeq2500 (Illumina, San Diego, CA, USA). Sequences obtained in the FASTQ files were passed through a quality filter of Q >30 and then de-multiplexed to obtain reads for each individual. Thereafter, the GBS reads were aligned to the International Wheat Genome Sequencing Consortium (IWGSC) Reference Sequence v1.0 (RefSeq v1.0) assembly and Rye Genome Sequencing Consortium Reference Sequence, using the Burrow-Wheelers Alignment tool (BWA) v0.7.4 [39]. Marker discovery, i.e., identification of SNPs, was accomplished using Samtools+Bcftools [40]. SNPs with minor allele frequency (MAF) <5% and more than 20% missing data were removed. After processing, 40,584 SNP markers for wheat and 15,116 SNP markers for chromosome 2R were retained for further analyses. Allele frequencies and genetic relationship between donor and recurrent parental lines were calculated using TASSEL v5.2.65 [41]. Principal component analysis (PCA) was performed using the function 'prcomp' in R 4.0.2. MapChart 2.2 (https://www.wur.nl/en/show/mapchart.htm) was used to draw physical maps.

## Selection of plants for recovery of recurrent parents

Progeny lines were selected based on their phenotypic and genomic similarity to the recurrent parents and used in the next backcross generation. The phenotypic parameters of each backcross generation ($BC_1F_1$ and $BC_2F_1$) were evaluated in the greenhouse, to determine whether the plants were similar to the recurring parents based on their height, tillering, heading date, flowering, spike characteristics (with or without awns), seed fertility, and maturity day. Background selection for alleles similar to those of the recurrent parents was then performed on the $BC_2F_2$ generation, using GBS markers distributed across all 42 wheat and the 2R rye chromosomes. Next, individual plants from each generation ($BC_2F_3$ to $BC_2F_6$) were carefully selected based on highest phenotypic and genotypic similarities to the recurrent parents, ensuring consistent inheritance of desired traits across generations.

$BC_2F_7$ and $BC_2F_8$ lines resistant to races TTTTF and TTKSK were sown on the field of Lantmännen Research Station in Svalöv (55.925621°N, 13.096742°E) for phenotyping evaluations. In one replicate field evaluation, these lines were sown in small plots to assess phenotypic traits compared with the recurrent parents. Data were collected on characteristics such as number of days to 50% flowering and maturity, plant height, tillering, lodging, susceptibility to diseases (e.g., rusts, powdery mildew, septoria, FHB, etc.), and grain color.

## Results

### Stem rust seedling response in the parental lines

Seventeen *Pgt* races were used to evaluate the seedling response of parental lines to stem rust. The results revealed that SLU238 and TA5094 were broadly resistant, exhibiting infection types (ITs) of; 1 to 1+2- to all races in this experiment (Table 1). The recurrent parents (BAJ#1, KACHU#1, and REEDLING#1) were found to be susceptible to several of the *Pgt* races and exhibited ITs of 3+4 to races TTTTF (USA and Russia), TTKTT, TTRTF, TTKSK, TTTSK, and TRTTF (Table 1). For races RKQQC, RCRSC, TKTTF (Sweden, Iraq, Ethiopia), LTBDC (Australian *Pgt* race 98–1,2,3,5,6), QTHJC, and TPMKC, recurrent parents had ITs ranging from 0 to 11+ (Table 1). CSph1bM and CSA were highly susceptible to all 17 *Pgt* races tested, indicating that no resistance genes were present in these two lines (Table 1). Based on the virulence profile of the different *Pgt* races, REEDLING#1, BAJ#1, and KACHU#1 were postulated to carry resistance genes *Sr11* and *Sr38* (Table 1).

### Stem rust seedling evaluations and marker-assisted backcrossing

Evaluation of the $BC_1F_1$ and $BC_2F_1$ populations using race TTTTF suggested presence of a major resistance gene following crossbreeding with BAJ#1 (Table 2). The P-values for all other $BC_1F_1$ and $BC_2F_1$ populations resulting from these crosses were <0.05, indicating a segregation pattern deviating from the expected 1:1 ratio. Such deviation suggests potential segregation distortion or the involvement of more than one major gene in resistance. However, the segregation ratio was close to 3:1 (P>0.05) for the $BC_2F_2$ populations from the backcrosses to BAJ#1 and REEDLING#1 (Table 2), indicating a single dominant major resistance gene. To verify presence of *Sr59* in plants that were selected for further generations, a number of resistant plants from each of the $BC_1F_1$, $BC_2F_1$, and $BC_2F_2$ populations were selected and genotyped with three KASP markers (Table 2). Plants found to contain the gene were transplanted for backcrossing and selfing (Table 2). The $BC_2F_2$ populations were also phenotyped with race TTKSK, which resulted in a significant deviation from the expected 3:1 segregation ratio (P<0.01). Instead, the segregation ratio was closer to 14:2 (P>0.05), suggesting presence of a major gene plus one or more additional genes co-acting with the major gene. The $BC_2F_4$ and

Table 1. Response to stem rust of seedlings from the parental lines used in this study.

| Parental line | TTTTF USA | TTRTF Italy | TTTTF Russia | TKTTF +Sr25 Iraq | TTKTT Rwanda | QTHJC USA | TPMKC USA | RKQQC USA | RCRSC USA | TKTTF Sweden | TKTTF Ethiopia | TTKSK Kenya | TTTSK Kenya | TTKST Kenya | JRCQC Ethiopia | TRTTF Yemen | LTBDC Australia | Gene postulation | Spike phenotype |
|---|---|---|---|---|---|---|---|---|---|---|---|---|---|---|---|---|---|---|---|
| CSph1bM | 4 | 4 | 4 | 4 | 4 | 4 | 3+ | 4 | 4 | 4 | 4 | 4 | 4 | 4 | 4 | 4 | 4 | None | Awnless |
| CSA | 4 | 4 | 4 | 4 | 4 | 4 | 3+ | 4 | 4 | 4 | 4 | 4 | 4 | 4 | 4 | 4 | 4 | None | Awnless |
| SLU238 | 1+2- | ;1 | 11+ | 11+ | ;1 | 11- | 11+ | 11+ | ;11- | 11+ | 11+ | ;1- | ;1- | ;1- | 11+ | ;11+ | 11+ | Sr59 | Awnless |
| TA5094 | 1+2- | ;1 | 11+ | 1+ | ;1 | 11- | 11+ | 11+ | 11+ | 11+ | 11+ | ;1- | ;1- | ;1- | 11+ | ;11+ | 11+ | Sr59 | Awnless |
| BAJ#1 | 4 | 4 | 4 | ;1- | 4 | ;0 | 11+ | ;0 | 11- | ;1- | 11+ | 4 | 4 | 4 | N.A. | 4 | 22- | Sr11+Sr38 | Awn |
| KACHU#1 | 4 | 4 | 4 | ;1- | 3+ | 11+ | 11- | ;0 | ;1- | ;1- | ;1- | 4 | 4 | 4 | N.A. | 4 | 11+ | Sr11+Sr38 | Awn |
| REEDLING#1 | 4 | 4 | 4 | ;1- | 4 | ;0 | 11- | ;0 | ;1- | ;1- | 11+ | 4 | 4 | 4 | N.A. | 4 | 11+ | Sr11+Sr38 | Awn |

N.A.–Not available. Infection types observed based on 0–4 scale [35]. Plants with infection types; 0 to 22- were considered resistant, while plants with infection types 3–4 were considered susceptible.

**Table 2. Crossing, backcrossing, and selection procedures with *Pgt* race TTTTF, KASP markers, and phenotyping selection in BC$_1$F$_1$ and BC$_2$F$_1$ populations.**

| Cross | Generation | *Pgt* race TTTTF | | χ2 | P-value | No. of plants for KASP analysis | No. of transplanted plants | Selected plants for GBS | Expected phenotypes for selection |
|---|---|---|---|---|---|---|---|---|---|
| | | Resistant | Susceptible | | | | | | |
| BAJ#1*1/TA5094 | BC$_1$F$_1$ | 265 | 195 | 10.65 | 0.001 | 94 | 40 | - | Awn and short height* |
| BAJ#1*2/TA5094 | BC$_2$F$_1$ | 135 | 155 | 1.37 | 0.24 | 94 | 40 | - | Awn and short height |
| BAJ#1*2/TA5094 | BC$_2$F$_2$ | 180 | 66 | 0.65 | >0.1 | 40 | 32 | 32 | Awn and short height |
| KACHU#1*1/ TA5094 | BC$_1$F$_1$ | 380 | 310 | 7.1 | 0.007 | 94 | 40 | - | Awn and short height |
| KACHU#1*2/ TA5094 | BC$_2$F$_1$ | 235 | 195 | 4.35 | 0.036 | 94 | 40 | - | Awn and short height |
| KACHU#1*2/ TA5094 | BC$_2$F$_2$ | 215 | 35 | 24.2 | <0.001 | 40 | 30 | 30 | Awn and short height |
| REEDLING#1*1/ TA5094 | BC$_1$F$_1$ | 520 | 400 | 15.65 | <0.001 | 94 | 40 | - | Awn and short height |
| REEDLING#1*2/ TA5094 | BC$_2$F$_1$ | 278 | 232 | 4.14 | 0.041 | 94 | 40 | - | Awn and short height |
| REEDLING#1*2/ TA5094 | BC$_2$F$_2$ | 432 | 127 | 2.32 | >0.1 | 80 | 66 | 66 | Awn and short height |

*Selection based on height ranging from 95 to 100 cm. Infection types observed based on 0–4 scale [35]. KASP markers were used to validate the presence of *Sr59* [16].

BC$_2$F$_5$ populations, obtained through assessments against race TTTTF and subjecting selected resistant plants to KASP genotyping, showed a similar high level of resistance (IT; 1- to; 11+) against races TTTSK, TTKST, and TRTTF as seen in their resistant parental lines (Table 1), indicating successful transfer of *Sr59* to the later generations. The BC$_2$F$_5$ families also displayed similar responses (IT; 01- to; 11+) as SLU238 and TA5094 when evaluated against races TPMKC, QTHJC, RKQQC, and RCRSC.

## GBS genotyping and physical location of *Sr59* on the 2R chromosome

Through alignment of raw GBS reads against the rye line 'Lo7' (International Rye Genome Sequencing Consortium (IRGSC)) reference genome), followed by filtering to remove SNPs with missing values ≤20% and minor allele frequency (MAF) ≥ 5%, the physical positions of the 15,116 SNPs obtained were mapped to chromosome 2R at 347,694 bp to 945,773,747 bp (Fig 1). In Fig 2B, the physical positions of the 15,116 SNPs from GBS (rye alleles) are shown in red color, whereas blue color indicates the presence of wheat alleles when mapping the BC$_2$F$_2$ populations. The physical positions of three KASP markers (c20194_115, c25837_157, and c21825_230) and the GBS results showed that *Sr59* was located in the 2RL segment (Fig 1). The three KASP markers were also used to track the *Sr59* introgression into the recurrent parent's background through the crossing scheme utilized in the present study. BLASTN searches of the three KASP markers against IRGSC positioned the *Sr59* resistance gene between 914,812,226 bp and 943,109,279 bp on chromosome 2RL (Fig 1). Presence of 2RL was also clearly verified in lines SLU238, SLU239, and TA5094, as demonstrated by the red color in Fig 2B, whereas 2D (wheat) was verified in BAJ#1, KACHU#1, REEDLING#1, CSA, CSph1bM, and susceptible lines (6-BAJ-S, 23-KACHU-S, 33-Reed-S, 35-Reed-S, and 45-Reed-S), as demonstrated by the blue color in Fig 2B. A strong association was observed between presence of the rye/wheat alleles determined by the GBS dataset and resistance (red color)/susceptibility (green color) reactions to race TTTTF and TTKSK (Fig 2A and 2C). Most of the BC$_2$F$_3$ lines showed homozygosity (100% red color) for resistance to the TTTTF race, although some lines segregated (40% red and 60% green; Fig 2A). The reaction to race TTKSK was

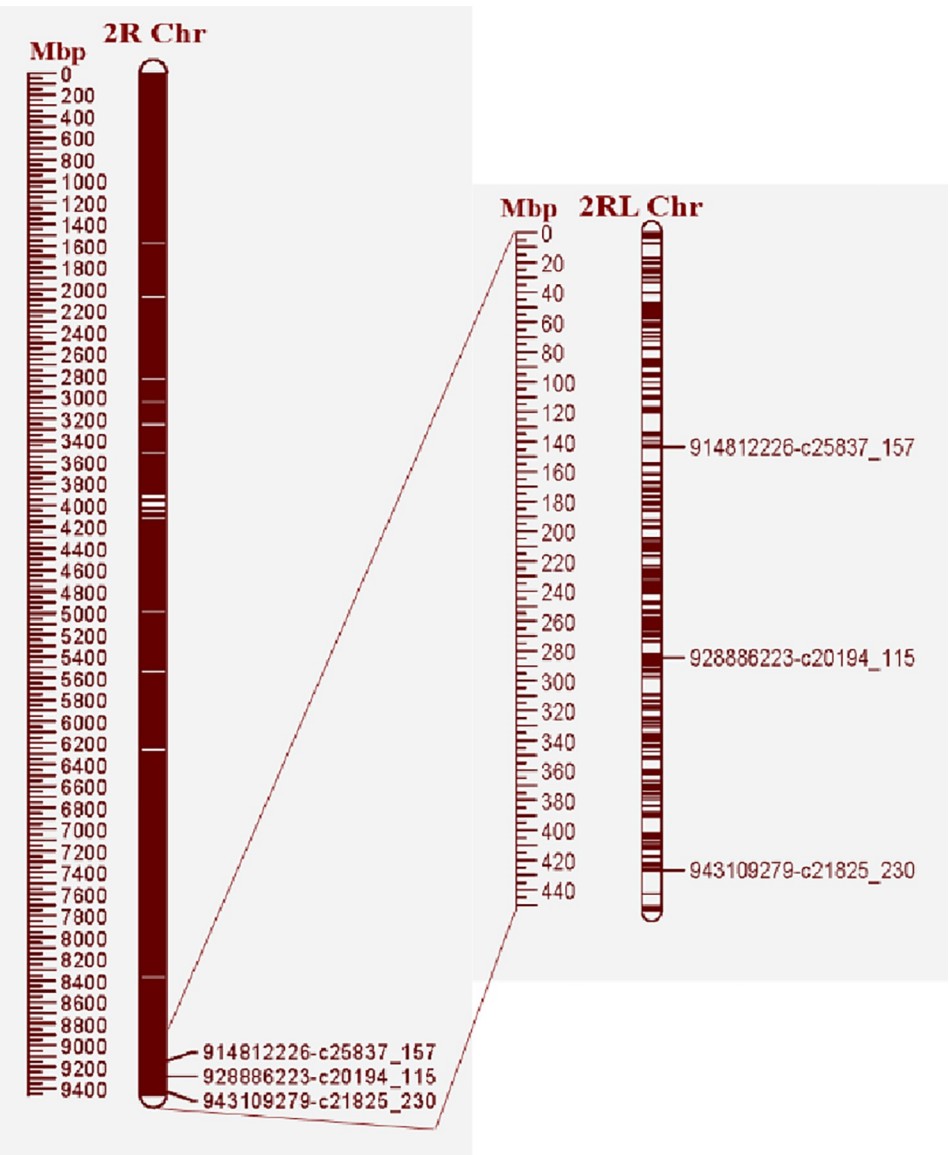

**Fig 1. Location of the *Sr59* resistance gene on rye chromosome 2R, determined using GBS reads aligned to the rye line 'Lo7'.** Dark red denotes presence of physical SNPs, while white denotes absence of SNPs throughout the 2R chromosome. The trio of SNPs on 2RL correspond to the three KASP markers identified previously.

tested in the $BC_2F_3$ lines, where the homozygous lines showed IT; 1 (100% red color), while the segregating lines showed; 1 to 3+4 (varying percentages of green and red color) (Fig 2C). The $BC_2F_3$ lines are shown as 100% green color for both races (TTTTF and TTKSK) in Fig 2A and 2C. PCA analysis based on the results of the GBS data (15,116 SNPs on chromosome 2R) clustered the genotypes evaluated into five distinct clusters: A) SLU238, SLU239, and TA5094; B) The $BC_2F_2$ population consists of recurrent parents harboring the 2RL chromosome proximate to SLU238 and TA5094; C) $BC_2F_2$ population derived from recurrent parents carrying the 2RL chromosome; D) Recurrent parents (BAJ #1, KACHU #1, and REEDLING #1) and the susceptible $BC_2F_2$ population without 2RL; and E) CSph1bM and CSA (Fig 3).

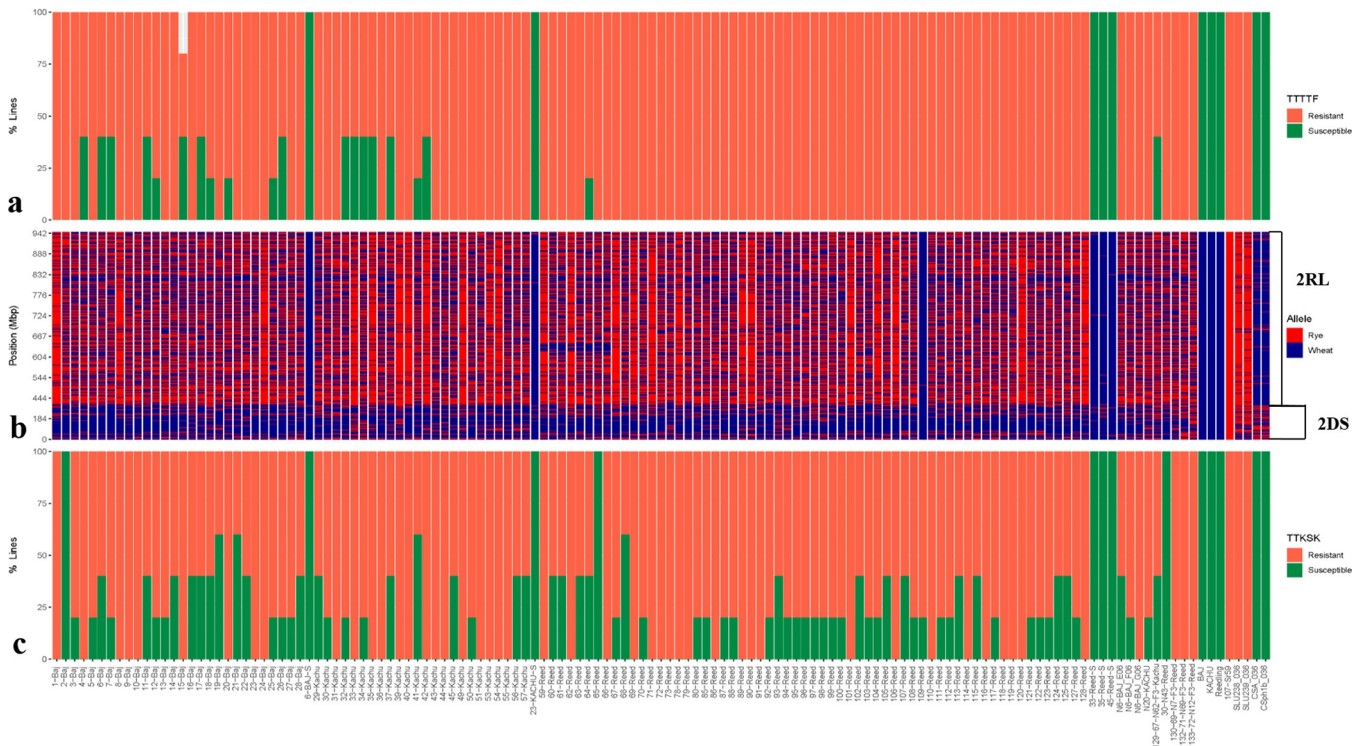

**Fig 2. Physical location of rye chromosome 2R based on GBS data and seedling responses to stem rust races TTTTF and TTKSK.** a) Seedling assay for race TTTTF in the $BC_2F_3$ population, where red denotes resistance and green susceptibility; b) physical positions of the 15,116 SNPs from GBS reads in the $BC_2F_2$ population, where red denotes the rye allele and dark blue the wheat allele; c) seedling assay for race TTKSK in the $BC_2F_3$ population, where red denotes resistance and green susceptibility.

### Recovery of the recurrent parent (background selection)

To select lines that resembled their recurrent parents as much as possible (with the exception of the addition of *Sr59*), a total of 40,584 SNPs across all 21 chromosomes were used to identify the most suitable lines in the $BC_2F_2$ population. Based on polymorphic SNPs, whole-wheat genome PCA identified five distinct clusters: 1) CSph1bM and CSA; 2) SLU238 and TA5094; 3) $BC_2F_2$ BAJ#1 population; 4) $BC_2F_2$ KACHU#1 population; and 5) $BC_2F_2$ REEDLING#1 population (Fig 4). The PCA results showed that most of the $BC_2F_2$ of a recurrent parent clustered at the same plot, indicating genomic recovery of the recurrent parent genome. As shown in Table 3, percentage genome recovery for the recurrent parents ranged from 66% to 75% across the three $BC_2F_2$ populations. The plants with the highest recurrent parent genome recovery and carrying *Sr59* were selected to generate homozygous lines through selfing.

### Phenotypic selection in greenhouse and field conditions

Besides using SNPs to produce lines resembling recurrent parental lines, selection was carried out in greenhouse and field evaluations with plant height and awns/awnless spikes being key phenotypic characters considered during crossing and backcrossing, as these characters differed between SLU238 and TA5094 compared with the recurrent parents, BAJ#1, KACHU#1, and REEDLING#1 (Table 2). Thus, 40 $BC_1F_1$ plants with maximum phenotypic similarity to the recurrent parents were used for developing the $BC_2F_1$ population. In $BC_2F_2$ to $BC_2F_6$ populations, plants were selected based on four characteristics: plant height, awns/awnless, days to maturity, and seed fertility. $BC_2F_7$ lines showing a homozygous resistance reaction to *Pgt* race

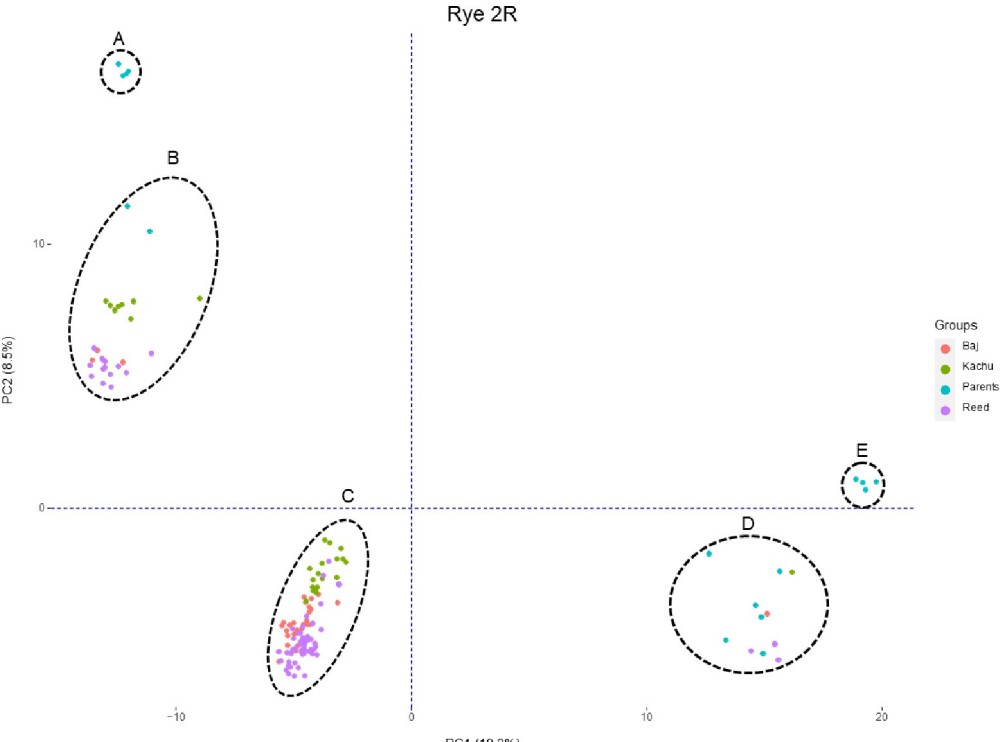

**Fig 3. PCA plot of rye chromosome 2R using 15,116 SNPs from GBS reads.** A) Resistant parental lines (SLU238, SLU239, and TA5094) with chromosome 2R; B) $BC_2F_2$ population comprising recurrent parents carrying chromosome 2RL close to SLU238 and TA5094; C) $BC_2F_2$ population derived from recurrent parents carrying the chromosome 2RL segment; D) recurrent parents (BAJ #1, KACHU #1, and REEDLING #1) and the susceptible $BC_2F_2$ population without chromosome 2RL; E) lines CSph1bM and CSA.

TTTTF and positive KASP marker data were sown in the field in 2020. The following phenotypic parameters were considered when selecting single plants in the field in 2020: plant stand, tillering, plant height, awns/awnless spikes, lodging, days to maturity, and seed fertility. Following the phenotypic analysis described above, $BC_2F_8$ populations were sown in the greenhouse and selfed to produce another generation, and $BC_2F_9$ populations were planted in the field in spring 2021 to select plants whose phenotypic similarity to the recurrent parents was greatest. The three KASP markers (c20194_115, c25837_157, and c21825_230) were used again to verify presence of the *Sr59* resistance gene in the individual $BC_2F_7$ to $BC_2F_9$ plants.

### *Ph1* allele status

A BLAST search against the IWGSC reference sequence v1.0 resulted in 2,050 GBS SNPs annotated in the range 16,637–712,890,017 bp on chromosome 5B (Fig 5). Furthermore, a deletion breakpoint of 50,936,209 bp (51 Mb), located from 396,630,846 bp to 447,567,055 bp, was detected in line CSph1bM, indicating the position of the *Ph1* locus on the 5B chromosome (Fig 5). The GBS data also revealed presence of the *Ph1b* deletion in TA5094 and two $BC_2F_2$ populations (31-Kachu and 34-Kachu), whereas it was not detected in the other $BC_2F_2$ populations, SLU238, CSA, or the recurrent parents. These results show that most of the $BC_2F_2$ populations carry the *Ph1* allele, and that the status of 2DS.2RL (and thus the introgression of *Sr59* into the wheat genetic background of the recurrent parents) is stable.

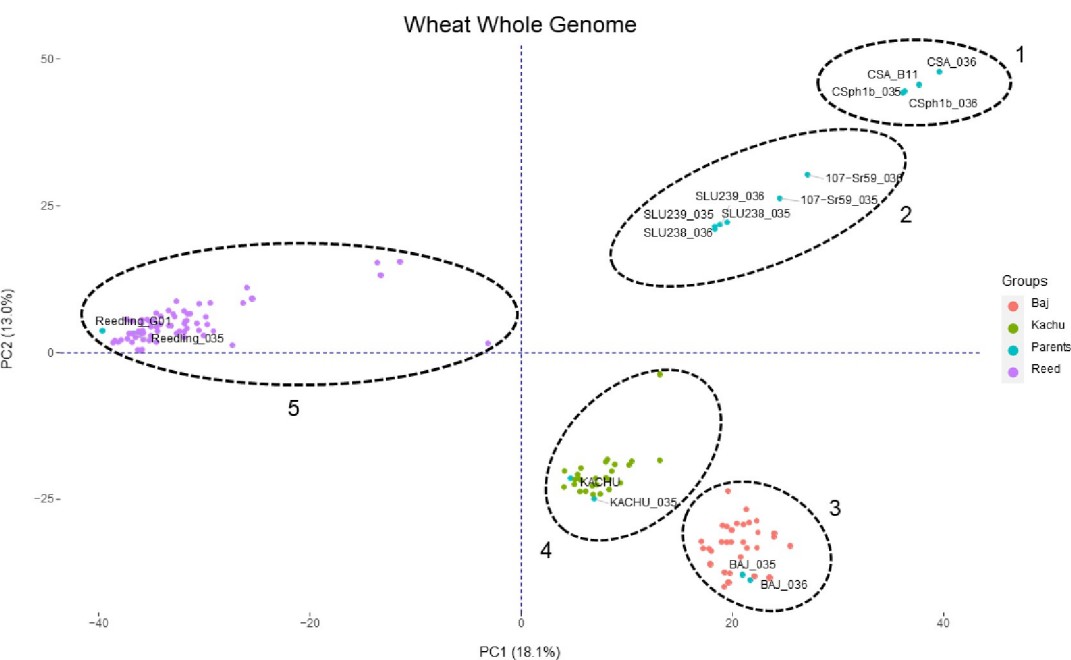

**Fig 4. PCA plot of wheat chromosomes using 40,584 SNPs from GBS reads.** Relationship between recurrent parents (BAJ#1, KACHU#1, and REEDLING#1) and lines SLU238, TA5094, CSph1bM, and CSA, based on polymorphic sites in the entire wheat genome as determined by GBS.

## Discussion

In this study, the *Sr59* stem rust resistance gene was transferred, using TA5094 as a donor of rye chromatin, into three elite wheat lines, through marker-assisted backcrossing selection and stem rust seedling screening ($BC_1F_1$ to $BC_2F_5$). Seedling screening and KASP marker analysis allowed us to trace presence/absence of *Sr59* from the parental lines in all progeny through all generations. The use of high-throughput genomic tools, such as GBS, facilitated the application of a strong selection pressure that increased the probability of recovering the elite recurrent parent background genome, while at the same time preserving the translocation fragment 2DS.2RL containing *Sr59*. The GBS background selection accurately identified both the translocated 2DS.2RL and the deletion region on 5BL (*ph1b* deletion). *Sr59* contributed stable resistance, as demonstrated by seedling screening against multiple *Pgt* races and KASP marker analysis across all populations developed.

Thus our novel GBS- and marker-assisted method was able to eliminate the *ph1b* deletion while transferring *Sr59* into the genetic background of three widely adapted wheat cultivars (BAJ#1, KACHU#1, and REEDLING#1) from CIMMYT through backcrossing, and can ultimately produce wheat lines suitable for breeders. Initially, the *Sr59* resistance gene was selected

**Table 3. Recurrent parent genome recovery in the $BC_2F_2$ generation using 40,584 genome-wide SNPs.**

| Cross | Generation | Genome recovered | | | Genome expected |
|---|---|---|---|---|---|
| | | Minimum | Maximum | Average | |
| BAJ#1*2/TA5094 | $BC_2F_2$ | 66.00% | 74.60% | 71.20% | 87.50% |
| KACHU#1*2/TA5094 | $BC_2F_2$ | 63.40% | 74.20% | 69.80% | 87.50% |
| REEDLING#1*2/TA5094 | $BC_2F_2$ | 66.2% | 73.40% | 70.50% | 87.50% |

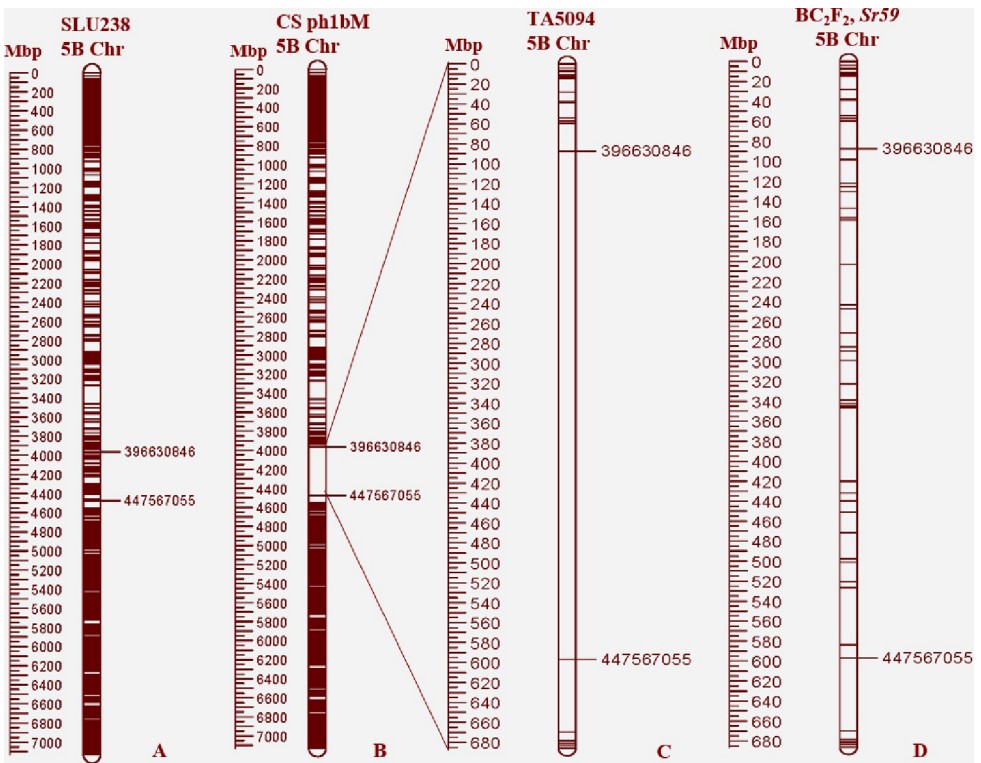

**Fig 5. Status of the *Ph1* allele (*Ph1* deletion), determined using GBS reads aligned to the IWGSC wheat reference sequence v1.0.** A) Line SLU238 depicting presence of the *Ph1* allele (SNPs) in chromosome 5B; B) deletion breakpoint in CSph1bM spans 50,936,209 bp, ranging from 396,630,846 bp to 447,567,055 bp; C) presence of the *Ph1b* deletion in TA5094 as revealed by GBS data; D) BC$_2$F$_2$ populations demonstrating stable presence of the *Ph1* allele.

based on the stem rust seedling response and marker-assisted backcrossing selection. Previous studies have shown that two backcrossing generations can recover approximately 87.5% of the recurrent parent genome [41]. Use of a large backcrossing population is common practice to introgress resistance genes from an alien genome into an elite background [25]. Aside from marker-assisted backcrossing, we used GBS genotyping to select plants with the greatest amount of the recurrent parent genome. Average recurrent parent genome recovery of 71.2%, 69.8%, and 70.5% was observed in the BC$_2$F$_2$ populations (Table 3). This fairly low recovery might be because of a less-than-optimal representation of 2DL in the GBS dataset, due to the fact that we started off with a 2DS.2RL translocation in the donor parent (Table 3). In GBS genotyping, the A and B genomes are reported to have the highest number of SNPs, while the D genome has the lowest number [42]. Evolutionary history and gene flow may be the reason for the poor D genome representation [43]. In previous studies, GBS has been found to be an inexpensive and robust approach for genotyping crop genomes, as it enables discovery of a high number of genome-wide markers, often SNPs [38]. Several studies have demonstrated that GBS detects small introgressions in wheat and barley [44, 45]. In the present, we study aligned GBS reads and located the physical positions of SNPs in wheat and 2R rye chromosomes based on the reference genome RefSeq v1.0 and the International Rye Genome Sequencing Consortium. In this alignment, GBS demonstrated the physical positions of 40,584 SNPs across the wheat genome and 15,116 SNPs for the 2R chromosome. Through this high-throughput genotyping procedure, it was possible to detect both translocation lines and non-translocation lines in the BC$_2$F$_2$ populations.

We also used PCA to visualize the grouping of lines based on their genetic relationship, i.e., based on the differences in 2RL segments transferred to the $BC_2F_2$ progeny (Fig 3). Basically, the susceptible $BC_2F_2$ plants and parental lines without chromosome 2R (i.e., CSph1bM, BAJ#1, KACHU#1, REEDLING#1) clustered with a positive PCA1 (Fig 3). Likewise, 24 $BC_2F_2$ resistant plants grouped closely to the lines TA5094 and SLU238, indicating presence of the whole 2RL chromosome segment (Fig 3). Additionally, a total of 28 $BC_2F_2$ resistant plants clustered differently, indicating that these lines most likely resemble each other as regards their genomic composition, for both wheat and rye genome segments (Fig 3). Some factors, such as chromosomal segment rearrangements, segmental duplications, and differences in recombination frequencies caused by genomic structural variations, may explain the marker orders observed in this short segment. The introgression of rye chromosomes into wheat genomes can result in structural changes and rearrangements, as the heterochromatin DNA of rye chromosomes can interfere with chromosome synapsis [2, 46]. There may have been chromosome rearrangements in the $BC_2F_2$ population that resulted in a shorter 2RL segment than in the other 24 $BC_2F_2$ lines. This study showed that the *Sr59* resistance gene in the 2RL chromosome segment is stable for normal transmission through the male gamete, preventing segregation distortion in cultivar development. No segregation distortion was observed in any of the three populations evaluated, either for race TTTTF or for race TTKSK. However, previous studies have shown that segregation distortion is a common feature when alien chromosomes are introgressed in the wheat genome [25]. Development of chromosome-specific SNP markers covering target chromosomes and facilitating homologous recombination on chromosomes containing resistance genes can assist in tracking rye resistance genes within wheat more effectively by minimizing chromosome transfer and reducing the likelihood of linkage to undesirable alleles. Line SLU238 wheat-rye disomic substitution carrying 2R (2D) chromosome exhibits effective resistance to several virulent races of stem rust and has been used to develop 2DS.2RL wheat-rye translocation lines [16]. Substitution lines serve as bridging materials in the development of wheat-alien translocation lines [25]. By incorporating distinct alien chromosome segments with desired traits through chromosome translocations, linkage drag can be reduced [25, 28].

For the three KASP markers (c20194_115, c25837_157, and c21825_230), BLASTN was used and their positions were mapped at 914,812,229 bp to 943,109,279 (28 Mb), as physically mapped in a previous study [16]. Based on the GBS reads, we observed a deletion spanning from 396,630,846 Mb to 447,567,055 Mb on chromosome 5B in TA5094 and two $BC_2F_2$ plants (31-Kachu and 34-Kachu), indicating deletion of the CSph1bM allele. A deletion on the 5B chromosome was not detected in the other $BC_2F_2$ plants, all carrying the *Ph1* allele, preventing true homology in pairing. New *ph1b* deletion-specific markers based on 90K SNPs have been developed to accurately identify the *ph1b* deletion region [47]. Successful cross-over between the wheat chromosome and its wild relatives is challenging, which is why CSph1bM mutants are recommended for inducing meiotic homoeologous recombination [23]. Several resistance genes, such as *Sr32*, *Sr47*, *Sr39*, *Sr59*, and *Yr83* have been successfully transferred using CSph1bM mutants [16, 24, 25, 29]. This approach provides an effective means of introducing beneficial traits from wild relatives for wheat improvement. The *Sr11* and *Sr38* resistance genes were postulated to be present in three recurrent parents from CIMMYT (Table 1). The *Sr11* and *Sr38* resistance genes have previously been reported to be widely prevalent in wheat cultivars worldwide [48, 49], and the recurrent parents may carry both genes. Due to the limited agronomic performance of the CSph1bM mutant and the elimination of the *ph1b* allele in line TA5094, it is necessary to transfer *Sr59* into adapted cultivars. Here, we successfully transferred *Sr59* from TA5094 to the genetic background of the three cultivars BAJ#1, KACHU#1, and REEDLING#1. The progeny showed proven resistance to TTTTF (USA), TKTTF

(Ethiopia), QTHJC, TPMKC, RKQQC, RCRSC, TTKST, TTKSK, TTTSK, TRTTF, and LTBDC (Australian *Pgt* race 98–1,2,3,5,6). Following backcrossing, morphological and agronomic characteristics with the greatest similarity to the recurrent parents were considered. Other resistance genes such as *SrTA1662*, *Yr15*, and *Sr39* have been transferred in previous work using backcrossing to recover the recurrent parent phenotype [25, 50, 51]. In conclusion, *Sr59* offers broad-spectrum resistance to stem rust races, making it a valuable gene for wheat improvement, while the reliability of KASP markers for *Sr59* makes them suitable for marker-assisted selection of stem rust resistance in wheat breeding. These findings can facilitate further production of stem rust-resistant wheat cultivars, developed with *Sr59* resistance in their elite background, which can act as additional assets for improving wheat yields and preventing stem rust losses.

## Acknowledgments

We thank the University of Minnesota Genomics Center for its genotyping services and the University of Minnesota Supercomputing Institute for providing computational resources. We also acknowledge Lantmännen for hosting the field trial at their Research Station in Svalöv.

## Author Contributions

**Conceptualization:** Matthew N. Rouse, Mahbubjon Rahmatov.

**Data curation:** Prabin Bajgain, Eva Johansson, Mahbubjon Rahmatov.

**Formal analysis:** Prabin Bajgain, Mahbubjon Rahmatov.

**Funding acquisition:** Eva Johansson, Mahbubjon Rahmatov.

**Investigation:** Mahboobeh Yazdani.

**Methodology:** Mahboobeh Yazdani, Mehran Patpour.

**Resources:** Mahboobeh Yazdani, Brian J. Steffenson, Mehran Patpour.

**Supervision:** Matthew N. Rouse, Brian J. Steffenson, Mahbubjon Rahmatov.

**Validation:** Matthew N. Rouse, Brian J. Steffenson, Prabin Bajgain, Eva Johansson, Mahbubjon Rahmatov.

**Visualization:** Matthew N. Rouse, Eva Johansson.

**Writing – original draft:** Mahbubjon Rahmatov.

**Writing – review & editing:** Mahboobeh Yazdani, Matthew N. Rouse, Brian J. Steffenson, Prabin Bajgain, Mehran Patpour, Eva Johansson, Mahbubjon Rahmatov.

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
