## [Decision Letter · Decision Letter 0]

31 Jul 2023

PONE-D-23-21095Developing adapted wheat lines with broad-spectrum resistance against stem rust: Introgression of Sr59 through backcrossing and selections based on genotyping-by-sequencing dataPLOS ONE

Dear Dr. Rahmatov,

Thank you for submitting your manuscript to PLOS ONE. After careful consideration, we feel that it has merit but does not fully meet PLOS ONE’s publication criteria as it currently stands. Therefore, we invite you to submit a revised version of the manuscript that addresses the points raised during the review process.

We look forward to receiving your revised manuscript.

Kind regards,

Pramod Prasad, Ph.D.

Academic Editor

PLOS ONE

Journal Requirements:

   "For this research financial support obtained from the Swedish Research Council Vetenskapsrådet and FORMAS "

Additional Editor Comments:

Please Revise the MS based on the suggestion of both the reviewers.

Thank you

Reviewers' comments:

Reviewer's Responses to Questions

**Comments to the Author**

1. Is the manuscript technically sound, and do the data support the conclusions?

Reviewer #1: Yes

Reviewer #2: Partly

2. Has the statistical analysis been performed appropriately and rigorously? 

Reviewer #1: Yes

Reviewer #2: I Don't Know

3. Have the authors made all data underlying the findings in their manuscript fully available?

Reviewer #1: Yes

Reviewer #2: Yes

4. Is the manuscript presented in an intelligible fashion and written in standard English?

Reviewer #1: Yes

Reviewer #2: No

5. Review Comments to the Author

Reviewer #1: The findings reported in this manuscript, along with the material that will be available, is valuable to the wheat research community in providing broad stem rust resistance due to the Sr59 allele that has been transferred into more adapted germplasm. Overall, the science is sound in this manuscript and ready for publication. Throughout, there are a few comments on some grammar mistakes, clarification on figures/tables, and also one little part of the materials and methods. Otherwise, the manuscript is really clear to read, and I congratulate the authors on doing a wonderful job on their research.

Major comments:

The abstract is too long. The limit listed for the journal is 300 words and the current abstract is over 400. I suggest finding a way to only include the most relevant findings for most people who will use these wheat lines. The information on the Ph1 locus could be eliminated from the abstract to reduce word count.

What was the field location used for phenotypic evaluations? Please provide a description, along with coordinates. Part of this research was done in Sweden, Denmark, and the US, which would greatly change the results for this part of the project. Also, please provide a more detailed description of small plots, such as rows, length, type, etc. Also, why was only one replication in the field used?

For all the figures and tables, not enough information is given for each. Please expand with more details for each one. For figure 1: which reference/genome are the physical positions based on, what are the dark red sections, and state that the markers are the three KASP markers. For figure 2: the panel description of a, b, and c doesn’t make what is in the figure so please correct, give a description of the colors used in the figure legend/text section, and more details on this figure. For figure 3, please describe each cluster A-E and make sure it matches what was in the text of the manuscript. For figure 4, please include cluster information. For figure 5, what is the reference used for all the physical positions? For table 1: what is the scale used or the citation for this, please remove the “e” from the end of “note” (I think you meant to say “not available”).

Minor comments:

Line 48: Include scientific name for wheat.

Line 52: Either take out “reached” or “attained”.

Line 54: Random “Wheat.” as its own sentence.

Line 58-59: This sentence doesn’t flow smoothly. It may sound better to rewrite as “…and may cause human health issues and be harmful to the environment [5].”

Line 62: remove “within” and rewrite sentence so has corrected grammar.

Line 71-72: Please fix grammar mistakes in this sentence as well.

Line 73: remove “about” before 35. Otherwise, state “about half”.

Line 115: Please state what type of cultivars these are. The thought behind this suggestion is so readers who are interested in requesting these lines can know what growth type (pretty sure all three are spring) and wheat species (Triticum aestivum).

Line 234: Please state which figure these KASP markers are on (pretty sure it is figure 1). The authors had switched briefly to discussing Figure 2b so it is confusing which one the KASP are on.

Lines 249-251: For figure 3, the clusters are labelled as letters in the figure, which don’t match the numbers in the text. Please change either the text or the figure labelling so they match.

Line 256-258 discusses clusters in Figure 4, but no clusters are labelled in this figure. Please label this figure so it matches the text.

Lines 312-315: Table 3 should be referenced in this section.

Line 334: There is no Figure 3B as stated in the text. Please correct to either figure 3 or the correct figure.

Line 372: Remove “was” after REEDLING#1.

Reviewer #2: TA5094 is the T2DS•2RL translocation line with the stem rust resistance gene Sr59. The present study described the transfer and subsequent evaluation of Sr59 into agronomically better genetic backgrounds using wheat cultivars as backcross parents. This is a necessary step for wheat breeding.

Some suggestions have been placed in the attached file.

A major concern is how to physically map the Sr59 resistance gene on chromosome 2RL (Figure 1). The authors should present a more detailed description.

6. PLOS authors have the option to publish the peer review history of their article (what does this mean?). If published, this will include your full peer review and any attached files.

Reviewer #1: No

Reviewer #2: No

---

## [Author Response · Author response to Decision Letter 0]

13 Sep 2023

Journal Requirements:

1. Please ensure that your manuscript meets PLOS ONE's style requirements, including those for file naming. The PLOS ONE style templates can be found at provided links.

Thank you for your suggestions and comments. We have adhered to the guidelines and styles provided by PLOS ONE at the given links.

The manuscript has been reviewed and edited by a professional language editor in the UK. Hopefully, this revision will now comply with PLOS ONE's language standards. 

Language editor: Mary McAfee. Email: mary.scantext@btinternet.com

3. Thank you for stating the following financial disclosure: "For this research financial support obtained from the Swedish Research Council Vetenskapsrådet and FORMAS"

There was no involvement of the funding agencies in the study's design, the collection and analysis of data, the decision to publish, or the preparation of the manuscript. We will write according to your suggestion. Thanks for your suggestions.

Review Comments to the Author

Reviewer #1

Suggestion by the Reviewer 1: The findings reported in this manuscript, along with the material that will be available, is valuable to the wheat research community in providing broad stem rust resistance due to the Sr59 allele that has been transferred into more adapted germplasm. Overall, the science is sound in this manuscript and ready for publication. Throughout, there are a few comments on some grammar mistakes, clarification on figures/tables, and also one little part of the materials and methods. Otherwise, the manuscript is really clear to read, and I congratulate the authors on doing a wonderful job on their research.

Response: Thanks for your positive suggestions and comments. 

Suggestion by the Reviewer 1: Major comments:

The abstract is too long. The limit listed for the journal is 300 words and the current abstract is over 400. I suggest finding a way to only include the most relevant findings for most people who will use these wheat lines. The information on the Ph1 locus could be eliminated from the abstract to reduce word count.

Response: We have revised the Abstract according to your recommendations and have now limited it to 298 words. Thank you for your suggestion.

Suggestion by the Reviewer 1: What was the field location used for phenotypic evaluations? Please provide a description, along with coordinates. Part of this research was done in Sweden, Denmark, and the US, which would greatly change the results for this part of the project. Also, please provide a more detailed description of small plots, such as rows, length, type, etc. Also, why was only one replication in the field used?

Response: The phenotypic evaluations took place in Svalöv, Sweden, and we have provided detailed descriptions of the field trials. Due to seed limitations and the use of BC2F7 and BC2F8 lines, where we noticed some segregation in certain families, we opted for a single replication. Seedling resistance screenings against stem rust races were performed at the Global Rust Reference Center at Aarhus University in Denmark, as well as the University of Minnesota and the United States Department of Agriculture's Agricultural Research Service Cereal Disease Laboratory in Minnesota, USA.

Suggestion by the Reviewer 1: For all the figures and tables, not enough information is given for each. Please expand with more details for each one.

Response: Thanks for your suggestion. We have revised the descriptions of the figures and tables in order to make them more informative.

Suggestion by the Reviewer 1: For figure 1: which reference/genome are the physical positions based on, what are the dark red sections, and state that the markers are the three KASP markers.

Response: Here, we utilized the genome assembly of the rye-inbred line 'Lo7' described by Rabanus-Wallace et al., 2021. We have also updated the figure's description.

Suggestion by the Reviewer 1: For figure 2: the panel description of a, b, and c doesn’t make what is in the figure so please correct, give a description of the colors used in the figure legend/text section, and more details on this figure.

Response: We have revised the descriptions of the figures in order to make it more informative.

Suggestion by the Reviewer 1: For figure 3, please describe each cluster A-E and make sure it matches what was in the text of the manuscript.

Response: Thank you for pointing it out. We have detailed the clusters A-E and verified that their descriptions are consistent with the text.

Suggestion by the Reviewer 1: For figure 4, please include cluster information.

Response: This change was incorporated in the manuscript revision.

Suggestion by the Reviewer 1: For figure 5, what is the reference used for all the physical positions?

Response: Here, we utilized the Wheat Genome Sequencing Consortium Reference Sequence v1.0 (RefSeq v1.0) for assembly. In the case of rye

Suggestion by the Reviewer 1: For table 1: what is the scale used or the citation for this, please remove the “e” from the end of “note” (I think you meant to say “not available”).

Response: We removed this accordingly

Suggestion by the Reviewer 1: Minor comments:

Line 48: Include scientific name for wheat.

Response: Triticum aestivum L. added in the text

Suggestion by the Reviewer 1: Line 52: Either take out “reached” or “attained”.

Response: Accordingly deleted

Suggestion by the Reviewer 1: Line 54: Random “Wheat.” as its own sentence.

Response: Wheat deleted

Suggestion by the Reviewer 1: Line 58-59: This sentence doesn’t flow smoothly. It may sound better to rewrite as “…and may cause human health issues and be harmful to the environment [5].”

Response: Here is a revised version of the sentence: Although fungicide application can effectively manage stem rust, it is associated with drawbacks such as high costs, significant environmental impact, and negative effects on human health

Suggestion by the Reviewer 1: Line 62: remove “within” and rewrite sentence so has corrected grammar.

Response: Here is a revised version of the sentence: An example of this is emergence of the Ug99 race group, which is capable of overcoming all known and extensively deployed stem rust (Sr) resistance genes, including Sr24, Sr31, Sr36, and SrTmp. This constant adaptation of the pathogen has increased concerns about global epidemics. Current line is 65 to 68.

Suggestion by the Reviewer 1: Line 71-72: Please fix grammar mistakes in this sentence as well.

Response: Here is a revised version of the sentence: Resistance genes such as Sr24, Sr31, and Sr38, present in German wheat cultivars, are limited in their effectiveness against these novel races of Pgt. Current line is 75 to 76.

Suggestion by the Reviewer 1: Line 73: remove “about” before 35. Otherwise, state “about half”.

Response: Accordingly removed

Suggestion by the Reviewer 1: Line 115: Please state what type of cultivars these are. The thought behind this suggestion is so readers who are interested in requesting these lines can know what growth type (pretty sure all three are spring) and wheat species (Triticum aestivum).

Response: Yes, they are all spring bread wheat. In the text “spring bread” is added.

Suggestion by the Reviewer 1:Line 234: Please state which figure these KASP markers are on (pretty sure it is figure 1). The authors had switched briefly to discussing Figure 2b so it is confusing which one the KASP are on.

Response: Thank you for bringing this issue to our attention. Figure 1 has been added

Suggestion by the Reviewer 1: Lines 249-251: For figure 3, the clusters are labelled as letters in the figure, which don’t match the numbers in the text. Please change either the text or the figure labelling so they match.

Response: We have updated the text and figure to reflect this change.

Suggestion by the Reviewer 1: Line 256-258 discusses clusters in Figure 4, but no clusters are labelled in this figure. Please label this figure so it matches the text.

Response: Accordingly, the cluster was placed as shown in the figure

Suggestion by the Reviewer 1: Lines 312-315: Table 3 should be referenced in this section.

Response: Table 3 is added as a reference to this section. Current line is 305 to 307.

Suggestion by the Reviewer 1: Line 334: There is no Figure 3B as stated in the text. Please correct to either figure 3 or the correct figure.

Response: The B is accordingly removed

Suggestion by the Reviewer 1: Line 372: Remove “was” after REEDLING#1.

Response: Accordingly removed

Reviewer #2

Suggestion by the Reviewer 2: TA5094 is the T2DS•2RL translocation line with the stem rust resistance gene Sr59. The present study described the transfer and subsequent evaluation of Sr59 into agronomically better genetic backgrounds using wheat cultivars as backcross parents. This is a necessary step for wheat breeding.

Response: Thanks for your positive suggestions and comments.

Suggestion by the Reviewer 2: Some suggestions have been placed in the attached file.

Response: Thank you for your suggestions on the manuscript. All of your suggestions have been appropriately incorporated into the manuscript.

Suggestion by the Reviewer 2: A major concern is how to physically map the Sr59 resistance gene on chromosome 2RL (Figure 1). The authors should present a more detailed description.

Response: We have revised the description of Figure 1, and we hope it now better explains Sr59's physical mapping.

Suggestion by the Reviewer 2: How to draw the conclusion according to the P-value in the BC1F1 and BC2F1.

Response: We calculated the P-value to assess the statistical significance of the observed impact of Sr59 on the TTTTF race, and here we draw conclusions based on this analysis.

---

## [Decision Letter · Decision Letter 1]

28 Sep 2023

Developing adapted wheat lines with broad-spectrum resistance to stem rust: Introgression of Sr59 through backcrossing and selections based on genotyping-by-sequencing data

PONE-D-23-21095R1

Dear Dr. Rahmatov,

We’re pleased to inform you that your manuscript has been judged scientifically suitable for publication and will be formally accepted for publication once it meets all outstanding technical requirements.

Kind regards,

Pramod Prasad, Ph.D.

Academic Editor

PLOS ONE

Additional Editor Comments (optional):

All the revision have been made.

Reviewers' comments:

Reviewer's Responses to Questions

**Comments to the Author**

1. If the authors have adequately addressed your comments raised in a previous round of review and you feel that this manuscript is now acceptable for publication, you may indicate that here to bypass the “Comments to the Author” section, enter your conflict of interest statement in the “Confidential to Editor” section, and submit your "Accept" recommendation.

Reviewer #2: All comments have been addressed

2. Is the manuscript technically sound, and do the data support the conclusions?

Reviewer #2: Yes

3. Has the statistical analysis been performed appropriately and rigorously? 

Reviewer #2: Yes

4. Have the authors made all data underlying the findings in their manuscript fully available?

Reviewer #2: (No Response)

5. Is the manuscript presented in an intelligible fashion and written in standard English?

Reviewer #2: Yes

6. Review Comments to the Author

Reviewer #2: The manuscript has been improved largely. The authors have adequately addressed my comments and I feel that this manuscript is now acceptable for publication.

7. PLOS authors have the option to publish the peer review history of their article (what does this mean?). If published, this will include your full peer review and any attached files.

Reviewer #2: No

---

## [Editor Report · Acceptance letter]

5 Oct 2023

PONE-D-23-21095R1 

Developing adapted wheat lines with broad-spectrum resistance to stem rust: Introgression of *Sr59* through backcrossing and selections based on genotyping-by-sequencing data 

Dear Dr. Rahmatov:

I'm pleased to inform you that your manuscript has been deemed suitable for publication in PLOS ONE. Congratulations! Your manuscript is now with our production department. 

Kind regards, 

on behalf of

Dr. Pramod Prasad 

Academic Editor

PLOS ONE